# Validation study of *HLA-B*13:01* as a biomarker of dapsone hypersensitivity syndrome in leprosy patients in Indonesia

Hana Krismawati[1], Astrid Irwanto[2,3,4], Arry Pongtiku[5], Ishak Darryl Irwan[2], Yustinus Maladan[1], Yuli Arisanti Sitanggang[1], Tri Wahyuni[1], Ratna Tanjung[1], Yonghu Sun[6], Hong Liu[6], Furen Zhang[6], Antonius Oktavian[1], Jianjun Liu[2,7]*

**1** Institute of Research and Development for Biomedicine Papua, National Institute of Health Research and Development, Jayapura, Indonesia, **2** Genome Institute of Singapore, Agency for Science, Technology and Research, Singapore, Singapore, **3** Department of Pharmacy, Faculty of Science, National University of Singapore, Singapore, Singapore, **4** Nalagenetics Pte Ltd, Singapore, Singapore, **5** Faculty of Public Health Medicine, University of Cendrawasih Papua, Jayapura, Indonesia, **6** Shandong Provincial Institute of Dermatology and Venereology & Provincial Hospital for Skin Diseases, Shandong First Medical University & Shandong Academy for Medical Sciences, Jinan, China, **7** Department of Medicine, Yong Loo Lin School of Medicine, National University of Singapore, Singapore, Singapore

☯ These authors contributed equally to this work.
* liuj3@gis.a-star.edu.sg

**Data Availability Statement:** The data containing the case/control status and each patient's HLA-B genotype is available in S1 Table.

## Abstract

Leprosy is a stigmatizing, chronic infection which degenerates the nervous system and often leads to incapacitation. Multi-drug therapy which consists of dapsone, rifampicin and clofazimine has been effective to combat this disease. In Indonesia, especially in Papua Island, leprosy is still a problem. Furthermore, there had been higher reports of Dapsone Hypersensitivity Syndrome (DHS) which also challenges leprosy elimination in certain aspects. Globally, DHS has a prevalence rate of 1.4% and a fatality rate up to 13%. The aim of this study is to validate *HLA-B*13:01*, a previously discovered biomarker for DHS in the Chinese population, as a biomarker for DHS in the Papua population. This is a case-control study of 34 leprosy patients who presented themselves with DHS (case subjects) and 52 leprosy patients without DHS (control subjects). Patients were recruited from 2 provinces: Papua and West Papua. DNA was extracted from 3 ml blood specimens. *HLA-B* alleles were typed using the gold-standard sequence based typing method. Results were then analysed using logistic regression and risk assessment was carried out. The results of HLA-typing showed that *HLA-B*13:01* was the most significant allele associated with DHS, with odds ratio = 233.64 and P-value = $7.11 \times 10^{-9}$, confirming the strong association of *HLA-B*13:01* to DHS in the Papua population. The sensitivity of this biomarker is 91.2% and specificity is 96.2%, with an area under the curve of 0.95. *HLA-B*13:01* is validated as a biomarker for DHS in leprosy patients in Papua, Indonesia, and can potentially be a good predictor of DHS to help prevent this condition in the future.

**Funding:** This work was supported by National Institute of Health Research and Development, Ministry of Health Republic of Indonesia (Grant no:2069.053.051 B, for authors HK, YM, YAS, TW, RT, AO), Genome Institute of Singapore, Agency for Science Technology and Research (Core Funding and Grant no: ETPL/17-GAP016-R20H, for authors AI, JL), National Key Research and Development Program of China (Grant no: 2016YFE0201500, for authors YS, HL, FZ). The funders had no role in study design, data collection and analysis, decision to publish, or preparation of the manuscript.

**Competing interests:** AI is an employee at Nalagenetics Pte Ltd. AI and JL have financial holdings at Nalagenetics Pte Ltd. All other authors have declared that no competing interests exist.

## Author summary

Leprosy as a chronic infectious disease that affects the skin and nervous system is treated with a treatment cocktail, including rifampicin, clofazimine and dapsone. Unfortunately, one of these drugs, namely dapsone, may cause the patient to exhibit adverse symptoms that appear as skin hypersensitivity and may potentially lead to death up to 9.9% of the time. In a previous study, it has been established that this adverse drug reaction, which is called dapsone hypersensitivity syndrome (DHS), is associated with an allele in the human leukocyte antigen, *HLA-B*13:01*. In the light of validating the association of *HLA-B*13:01* with DHS in a leprosy endemic area like Indonesia, we conducted a study involving leprosy patients who had DHS during the course of their multi-drug treatment as well as leprosy patients who managed to complete their course of treatment without exhibiting DHS. The results of this study validated the association of *HLA-B*13:01* and DHS at a very significant level of evidence and the odds of people carrying at least one allele is 233.3-times the risk of non-carriers. This allele was also able to predict a person at risk of DHS 95% of the time. Thus, screening of *HLA-B*13:01* to prevent DHS is warranted.

## Introduction

In the era of leprosy elimination, many efforts have been done by the government to combat leprosy[1]. Multi-drug treatment (MDT) supplied by the World Health Organization is intensively applied as the main treatment of leprosy in every region in Indonesia[2]. Although the availability of data is scarce, many cases of Dapsone Hypersensitivity Syndrome (DHS) were reported from districts in Papua and oftentimes are fatal cases[3,4].

Dapsone is an antibiotic and anti-inflammatory agent used for the prevention and treatment of infectious and chronic inflammatory diseases. As an antibiotic, dapsone is used to treat leprosy, malaria, actinomycetoma as well as *Pneumocystis jirovecii* pneumonia in persons with human immunodeficiency virus (HIV), whereas as an anti-inflammatory agent, it's used to treat dermatitis herpetiformis, linear IgA dermatosis, subcorneal pustular dermatosis and erythema elevatum dilutinum[5,6]. Among all these diseases, leprosy is currently the one that most prevalently uses dapsone for treatment.

About 0.5–3.6% of persons treated with dapsone have drug hypersensitivity syndrome[6–8], which was first described by Lowe and Smith[9] in 1949 and termed "dapsone hypersensitivity syndrome" (DHS)[10]. The syndrome is a severe idiosyncratic drug reaction characterized by the clinical manifestations of fever, rash, and systemic involvement (most commonly of the liver and the hematologic system), which can cause severe organ dysfunction. DHS is usually manifested 4 to 6 weeks after the initiation of therapy[11].

With the introduction of MDT for leprosy worldwide and with the use of dapsone in chemoprophylaxis for *P. jirovecii* pneumonia in HIV infection combined with trimethoprim/sulfamethoxazole, the incidence of DHS might have also increased. On the basis of a recent systematic review of the published epidemiologic studies, the estimated global prevalence of the DHS is 1.4%, and the associated mortality is 9.9%[12]. This review also showed that 72.5% of the global DHS incidence actually originated in Asia and 71.8% of these DHS cases happened in leprosy patients. A recent literature review for DHS in China concluded a prevalence of 1.5% and mortality rate of 9.6%, very close to the global estimate for DHS[13].

Previous study on the genetic determinant of DHS in 2,141 Chinese leprosy samples has discovered the risk factor for DHS[14]. They found that a single DNA variant, located in an immunity-associated gene called *HLA-B*, was significantly more common in those who

developed DHS. Individuals with one copy of *HLA-B*13:01* risk variant are 34-times more likely to develop DHS than individuals without. Two copies of *HLA-B*13:01* makes individuals 101-times more susceptible to the syndrome. The results have been independently validated in various Asian populations including Thais[15], Koreans[16], Indians[17] and Taiwanese[18]. Screening for this allele in new leprosy patients before the initiation of MDT has proven beneficial to reduce incidence rate of DHS in China to zero[19].

In this study we aim to validate *HLA-B*13:01* as a biomarker for DHS in the Indonesian population, report the allele frequency of *HLA-B*13:01* in Indonesia, and confirm the sensitivity of *HLA-B*13:01* as a predictor of DHS in this population.

## Methods

### Study design

This is a case-control genetic association study that collected retrospective leprosy patients who had survived DHS ('cases') and those whom did not exhibit DHS but has completed at least 6 months of multi-drug treatment for leprosy ('controls'). DHS cases in this study have all been free from DHS and had completed their MDT course excluding dapsone without any adverse reactions.

### Study population

The study was carried out in main pockets in the districts within Papua (Jayapura and Biak) and West Papua (Bintuni, Manokwari and Sorong) provinces in Indonesia.

### Ethical clearance

Ethical approval was obtained from National Institute of Health Research and Development with reference number LB. 02.01/5.2/KE.065/2016.

### Subject recruitment

The patients were recruited by inviting them to the point of care or visiting their homes. The informed consent was explained and given to the patient to read. The information sheet attached to the consent form contained the study aim, participant's eligibility, risks and benefits of participation, as well as confidentiality of data. The patients who agreed to participate in the study signed the inform consent in front of a researcher, a health worker and one family member as witnesses.

### Subject recruitment criteria

Leprosy patients treated with MDT who developed DHS symptoms were designated as 'cases'. The criteria to determine 'case' subjects followed what was described by Richardus and Smith [11] and had been translated into Bahasa Indonesia in a local guideline published by the Health Authority in Papua Province and Netherlands Leprosy Relief Papua[20]. These patients exhibit symptoms which include: (1) presence of at least two symptoms: fever, skin eruption, lymphadenopathy and liver abnormalities (elevated transaminases by at least 2-times); (2) symptoms appearing between the second and eighth week after the commencement of dapsone and disappearing upon discontinuation of the drug; (3) symptoms not ascribed to any other drug given simultaneously; (4) symptoms not attributable to leprosy reactions; (5) no other diseases liable to cause similar symptoms. Leprosy patients treated with MDT who did not develop any hypersensitivity reactions after 6–24 months of MDT treatment was designated as 'controls'. Each case has at least one control obtained from the same area. Subjects who have any

comorbidities or are under other types of medication other than the standard MDT were excluded from the study. The clinical information of the patients as well as the demographic and geographic origin were obtained. We took only one patient in each family based on their family name to ensure that all the samples were independent of each other and the genetic results we see are not due to familial segregation. The final number of samples that were collected are 34 cases and 55 controls, i.e. 14 cases and 31 controls from Papua province and 20 cases and 24 controls from West Papua province.

### DNA extraction and quantification

3 ml of EDTA anticoagulated venous blood was collected from all participants. The blood tubes were stored at -20˚C and transferred in chilled (4˚C) condition to Institute of Research and Development for Biomedicine Papua (IRDBP). At IRDBP, genomic DNA was extracted from the blood lymphocytes by a standard procedure using QIAGEN QIAamp DNA blood maxi kit (Catalog No. 51194). We modified the procedure from the manual to get the optimum yield, i.e. prolonged the incubation period at 70˚C for 2 hours instead of 10 minutes. The DNA yield was quantified using Nanodrop™ Spectrophotometry.

### HLA sequence based typing (SBT)

All of the samples were typed using a research-use-only (RUO) version of an approved (CE-IVD) HLA-typing kit commercially available from TBG Diagnostics Limited, which is called HLAssure SE B Locus SBT Kit (Catalog No. 50210). This kit is a direct sequence-based typing approach, which is done by capillary sequencing and considered as gold-standard for HLA typing. The kit is designed to detect all known polymorphic positions in exons 2 and 3 of *HLA-B*, which are necessary for accurate allelic determination. We followed the manufacturer's instructions for the HLA-typing experiment protocol. Results from the sequencing is processed through a licensed software that comes with the kit, called AccuType™ Software. Analysis procedure and interpretation followed the software's instruction manual.

### Power calculation

We assessed power using QUANTO[21] for the validation of the *HLA-B*13:01*, assuming a similar odds ratio (OR) with the initial discovery in Chinese[14] applies to the Indonesians (Fig 1). Assuming the frequency of *HLA-B*13:01* in Indonesians is around 2% (similar to the Chinese), the sample size to analyze in this proposed study will have a good power ($> 0.8$) to capture the associations of *HLA-B*13:01* at a type-1 error rate of 5% if we have at least 15 cases and 15 controls. Therefore, our collection of 34 cases and 55 controls are well powered to detect a positive association.

### Statistical analysis

Data analysis was done using R, which is a dedicated statistical and data management tool for genetic analysis. The statistical difference between the age distribution in cases and controls was calculated based on Student's t-test, while gender and geographical distribution was calculated based on Pearson's chi-square test. Association test of *HLA-B*13:01* and DHS was done by performing logistic regression. Statistical significance is defined as $P < 0.0016$ after considering Bonferroni correction for multiple testing of 30 *HLA-B* alleles. Receiver-operating curve was drawn using "pROC" package in R. Full data table containing *HLA-B* haplotypes of each sample as well as information on covariates and affection status is provided in S1 Table.

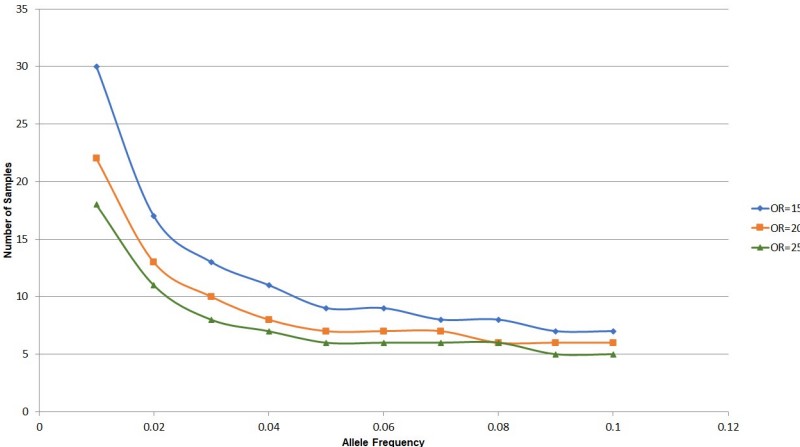

**Fig 1. Power estimated for various allele frequencies and effect sizes.**

### Interpretation of results

P-value was interpreted as the statistical significance of the test results as compared to by chance alone. A small p-value ($<0.0016$) indicates evidence against the null hypothesis of no association, hence it will be rejected, whereas a large p-value ($>0.0016$) indicates weak evidence against the null hypothesis, hence we fail to reject it. Odds ratio (OR) is a relative measure of effect comparing the DHS cases relative to the non-DHS cases. If the OR equals to 1, it implies no difference between the two groups.

## Results

In this study we recruited 34 leprosy patients with DHS as cases and 55 leprosy patients without DHS as controls. As the study subjects were recruited with a uniform criterion and matched according to the geographic region, the difference between the distribution of gender and age in cases and controls were not statistically significant (P>0.05, Table 1). This minimizes the chances of the genetic association results being attributable to artefacts from baseline characteristics of the study subjects.

Before genotyping, one control that failed to have a good quality DNA (yellow extract) with a 260/280 OD ratio of 1.31 was excluded. Excluding this sample, we had 34 cases and 54 controls typed on the HLAssure B Locus kit. Unfortunately, two other controls were not able to be sequenced successfully as only 25% of the reads were able to be mapped on the AccuType™ Software, and it failed to distinguish which *HLA-B* alleles these sample carry. Finally, excluding

**Table 1. Baseline characteristics of cases and controls.**

| Characteristic | Controls (n = 52) | Cases (n = 34) | Total (n = 86) | P-value |
|---|---|---|---|---|
| Age (mean ± SD) | 25.3 ± 10.1 | 25.0 ± 11.1 | 25.1 ± 10.5 | 0.900 |
| Gender (n; %) | | | | |
| Male | 31 (59.6) | 23 (67.6) | 54 (62.8) | 0.450 |
| Female | 21 (40.4) | 11 (32.4) | 32 (37.2) | |
| Male/Female Ratio | 1.47 | 2.09 | N/A | |
| Geographical Region (n; %) | | | | |
| Papua | 24 (46.2) | 20 (58.8) | 44 (51.2) | 0.251 |
| West Papua | 28 (53.8) | 14 (41.2) | 42 (48.8) | |

all the failed samples, 34 cases and 52 controls with successful *HLA-B* typing results were brought for further analyses.

## HLA-B*13:01 is the most associated allele with DHS in Papuans

Based on the result of the SBT, we can see a spectrum of *HLA-B* allele frequency in the leprosy patients from Papua (Table 2). We performed a log additive test (logistic regression) to observe the association of all the identified *HLA-B* alleles to DHS, but we discovered that only *HLA-B*13:01* allele showed association with DHS. *HLA-B*13:01* is present in 50% of the cases, 1.9% of the controls and the difference between the two groups is significant (P = $7.11 \times 10^{-9}$; odds ratio = 233.64), surpassing the genome-wide significance threshold. When the region where the samples came from was considered as a potential confounding factor (Papua or West Papua province), the adjusted P-value and odds-ratio of *HLA-B*13:01* is considered

**Table 2. Allele frequency spectrum of Papua population.**

| *HLA-B* alleles | AF controls (n = 52) | AF cases (n = 34) | AF ALL (n = 86) | P-value (unadjusted) | OR (95% CI) (unadjusted) | P-value (adjusted for covariates*) | OR (95% CI) (adjusted for covariates) |
|---|---|---|---|---|---|---|---|
| **13:01** | **0.019** | **0.5** | **0.209** | **$7.11 \times 10^{-9}$** | **233.64 (1.69–67.7)** | **$1.32 \times 10^{-7}$** | **328.87 (1.44–106.7)** |
| 15:02 | 0 | 0.015 | 0.006 | 0.991 | | | |
| 15:06 | 0.094 | 0 | 0.058 | 0.084 | | | |
| 15:12 | 0.009 | 0 | 0.006 | 0.991 | | | |
| 15:21 | 0.160 | 0.061 | 0.122 | 0.150 | | | |
| 15:25 | 0 | 0.015 | 0.006 | 0.992 | | | |
| 15:195 | 0.009 | 0 | 0.006 | 0.990 | | | |
| 18:01 | 0.160 | 0.091 | 0.134 | 0.099 | | | |
| 18:02 | 0 | 0.015 | 0.006 | 0.991 | | | |
| 18:105 | 0 | 0.015 | 0.006 | 0.991 | | | |
| 27:04 | 0.038 | 0 | 0.023 | 0.989 | | | |
| 27:06 | 0.019 | 0 | 0.012 | 0.992 | | | |
| 35:01 | 0 | 0.015 | 0.006 | 0.991 | | | |
| 35:05 | 0.038 | 0.015 | 0.029 | 0.404 | | | |
| 35:30 | 0.009 | 0 | 0.006 | 0.992 | | | |
| 38:02 | 0.132 | 0.030 | 0.093 | 0.046 | | | |
| 38:46 | 0 | 0.015 | 0.006 | 0.991 | | | |
| 39:01 | 0 | 0.015 | 0.006 | 0.991 | | | |
| 40:01 | 0.057 | 0.076 | 0.064 | 0.893 | | | |
| 40:02 | 0.028 | 0 | 0.017 | 0.991 | | | |
| 40:10 | 0.009 | 0.030 | 0.017 | 0.323 | | | |
| 40:11 | 0.009 | 0 | 0.006 | 0.992 | | | |
| 40:89 | 0.009 | 0 | 0.006 | 0.992 | | | |
| 40:99 | 0.009 | 0 | 0.006 | 0.992 | | | |
| 40:143 | 0.009 | 0 | 0.006 | 0.992 | | | |
| 48:01 | 0.009 | 0.061 | 0.029 | 0.084 | | | |
| 56:01 | 0.066 | 0 | 0.041 | 0.991 | | | |
| 56:02 | 0.009 | 0 | 0.006 | 0.992 | | | |
| 56:07 | 0.075 | 0.030 | 0.058 | 0.224 | | | |
| 58:59 | 0.009 | 0 | 0.006 | 0.992 | | | |

*covariates include age, gender, geographical region.

AF, allele frequency. Considering Bonferroni correction for testing of 30 *HLA-B* alleles, the P-value for significance is *P*<0.0016.

**Table 3. Allele and genotype frequencies of *HLA-B*\*13:01** and the Hardy Weinberg P-Values in cases and controls.**

| *HLA-B*\*13:01 | Non-carrier | Carriers | | Total carriers | Allele frequency | HWE P-value |
|---|---|---|---|---|---|---|
| | | Heterozygous | Homozygous | | | |
| **DHS +** (N = 34) | 3 (8.82%) | 28 (82.35%) | 3 (8.82%) | 31 (91.17%) | 50% | 0.00039 |
| **DHS—**(N = 52) | 50 (96.15%) | 2 (3.85%) | 0 (0%) | 2 (3.85%) | 1.92% | 1 |

unaffected (adjusted P = $1.08 \times 10^{-8}$; odds ratio = 241.68), indicating that the evidence of association was not driven by the effect of population stratification. Additionally, we also tested for the confounding of age, gender and region altogether and found that the P-value remain highly significant (adjusted P = $1.32 \times 10^{-7}$; odds ratio = 328.87).

### *HLA-B*\*13:01 as a risk predictor for DHS in Papuans

There were 31 carriers and 3 non-carriers of *HLA-B*\*13:01 among the DHS cases; whereas among the controls, there were 2 carriers and 50 non-carriers (Table 3). In other words, *HLA-B*\*13:01 was present in 91.2% of case patients and 3.8% of the controls. This suggests that *HLA-B*\*13:01 had a sensitivity of 91.2% and specificity of 96.2% as a predictor of DHS. However, we also observed a significant deviation of the genotype frequency of *HLA-B*\*13:01 in the case patients (P = 0.00039), unlike the controls (P = 1).

Using *HLA-B*\*13:01 as a predictor of DHS risk in Papuans, we observed that the area under the curve (AUC) = 0.95, which means that *HLA-B*\*13:01 can correctly classify people with and without risk of developing DHS 95% of the time (Fig 2). Consequently, the risk carried by a

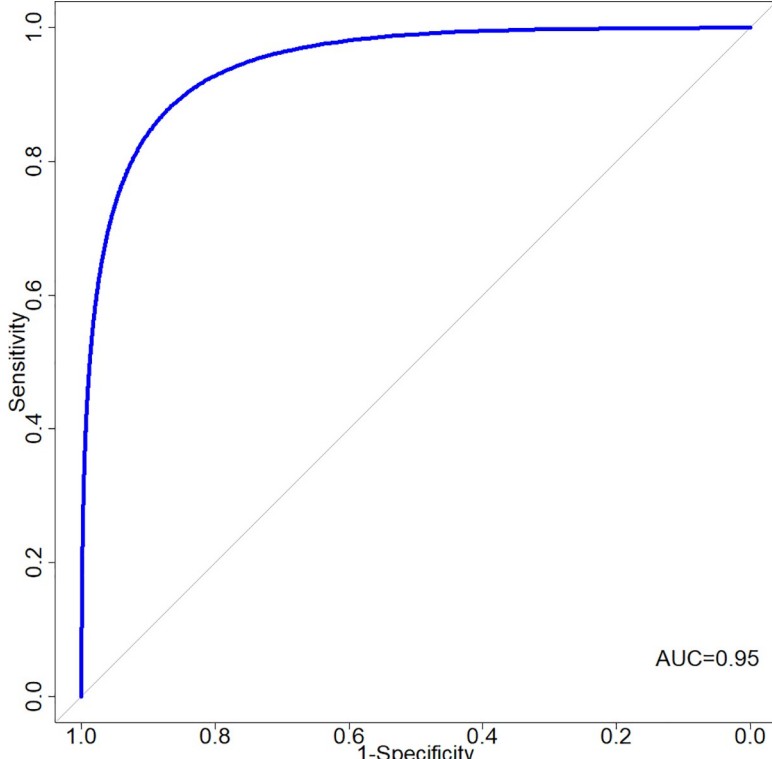

**Fig 2. Receiver-operating curve for an additive prediction model of DHS using *HLA-B*\*13:01 allele as predictor in Papuans.** AUC, area under the curve. Overall sensitivity is 91.2% and specificity is 96.2%.

person who has at least one copy of *HLA-B*13:01* allele (heterozygous odds ratio) is 233.3-times higher than the risk of those who do not carry any copy of *HLA-B*13:01* allele. Assuming that the prevalence of DHS in Papua is similar to the estimated global prevalence of 1.4%[12], *HLA-B*13:01* would have a positive predictive value (PPV) of 25.4% and a negative predictive value (NPV) of 99.9% in identifying people at risk to DHS. To prevent one case of DHS, we would theoretically need to screen 79 leprosy patients and hence we could reduce DHS risk by 10-fold (from 1.4% to 0.13%). Our data from 2016 in Papua has shown that the incidence of DHS among leprosy patients is 10-times higher than the global estimate (~11%). This translates to a PPV of 74.8%, NPV of 98.9%, and 10 people needed to screen to prevent one case of DHS.

## Discussion

In this study, we replicated the association of dapsone hypersensitivity syndrome in the Indonesian population, where Papua and West Papua provinces were selected as they are considered top three largest leprosy pockets in Indonesia with a prevalence of 4.06 per 10,000 population in Papua and 11.48 per 10,000 population in West Papua. These two areas are also considered as focus community areas in the National Program for Leprosy Elimination. In terms of DHS, according to an unpublished annual local report, Papua has an incidence between 2–10% per year. Besides, Papua New Guinea, the neighboring country sharing the same island as Papua has shown to have the highest reported allele frequency in the world (28%), unlike other major ethnicities in Indonesia, such as Javanese and Sundanese (1.5%) and Riau population (5.4%)[22].

Initially discovered as a risk factor and predictor of DHS in the Chinese population[14], *HLA-B*13:01* has been shown in multiple other studies in Asia as being a strong predictor to the same condition[15–18,23]. It has also been shown to be associated with DHS occurring in non-leprosy patients in Thailand[15]. Compared to the studies in Chinese and Thais, the odds ratio we discovered in Papuans are 11-times what was reported in Chinese[14] and 4.3-times what was reported in Thais[15]. Presumably, the abundance of *HLA-B*13:01* in Papuans contribute to the high odds ratio we see in this population. In Chinese, the frequency of *HLA-B*13:01* ranges between 3–8% across North and South China, but for the Melanesians and Australian Aboriginals who are evolutionary related to Papuans, the allele frequency may reach up to 28%[22].

Similar to the publication in the Chinese population[14], we found a significant deviation of *HLA-B*13:01* genotype frequency from Hardy Weinberg equilibrium in only the case patients (P = $3.9{\times}10^{-4}$). We also think the reason for this is due to the potential oversampling as the effect of the highly penetrant *HLA-B*13:01* as a susceptibility factor for DHS and the bias towards sampling only the surviving DHS patients. Although not very well documented, the mortality rate for DHS in Papua is estimated to be quite high. Many of the patients come from rural areas and oftentimes they were treated after active case findings into their home villages. Upon drug delivery, any side effect that occur to them were often not reported. Patients even think these side effects were curses instead of a medical condition. Hence, with not much understanding of the disease and potential side effects, many of the fatal cases would have been missed and being under-reported.

Until now, there is no preventive action to reduce DHS incidence that been implemented in hospitals as well as in point of care clinics to prevent DHS in Indonesia. The DHS treatment is performed with intensive care at the time DHS symptoms appear, and it may be costly as the patient may require hospitalization for several days up to weeks. Besides the lack of knowledge and public awareness of DHS, Indonesia's geographic landscape adds as a challenge for

patients who need immediate treatment upon a severe adverse reaction. There have been reported deaths in DHS patients due to delays in handling and some community perceptions that stigmatizes DHS as a curse instead of a form of an adverse drug effect. The incidence of DHS case in the family or community also causes trauma as well as rejection to undergo MDT. This issue affects leprosy elimination program especially in endemic areas of leprosy in Indonesia.

Screening of *HLA-B*13:01* among leprosy patients before treatment can be valuable in many aspects: clinically to prevent DHS and its related mortality, socially to prevent trauma of losing family members and experiencing severe hypersensitivity, and in public health perspective to increase compliance to treatment and reduce transmission for reaching the goal of national elimination program. We believe that by implementation of a cost-effective genomic test, health programs can prevent DHS occurrence close to zero, thereby increasing MDT's safety and confidence as a standard of care, while demonstrating cost savings to the health care system and relieving an enormous burden on quality of life and potential death on patients and their families.

## Limitation

As a case-control population-based study, participants were recruited based on past history of DHS diagnosis as written in the medical record. The limitation to this is that misdiagnosis of DHS may have happened and there will be no way to verify this real-time. However, to alleviate this issue, during sample collection we visited the subjects and conducted in-depth interviews accompanied by the attending physician and/or medical doctor who diagnosed and treated the subject when DHS had manifested.

## Conclusion

*HLA-B*13:01* is validated as a biomarker for DHS in leprosy patients in Indonesia. Screening for *HLA-B*13:01* may potentially reduce DHS incidence significantly and benefit many aspects of the leprosy control program in Indonesia.

## Supporting information

**S1 Table. Full data table containing HLA-B haplotypes of each sample as well as information on covariates and affection status.**
(XLSX)

## Acknowledgments

We thank all the individuals who had participated in this project and the healthcare workers who had accompanied in the sample and data collection.

## Author Contributions

**Conceptualization:** Astrid Irwanto, Jianjun Liu.

**Data curation:** Arry Pongtiku, Yustinus Maladan, Yuli Arisanti Sitanggang, Tri Wahyuni, Ratna Tanjung.

**Formal analysis:** Hana Krismawati, Astrid Irwanto.

**Funding acquisition:** Astrid Irwanto, Furen Zhang, Jianjun Liu.

**Investigation:** Hana Krismawati, Arry Pongtiku, Ishak Darryl Irwan, Antonius Oktavian.

**Methodology:** Astrid Irwanto, Arry Pongtiku, Yustinus Maladan, Yuli Arisanti Sitanggang, Tri Wahyuni, Ratna Tanjung.

**Project administration:** Hana Krismawati, Ishak Darryl Irwan, Yustinus Maladan, Yuli Arisanti Sitanggang, Tri Wahyuni, Ratna Tanjung.

**Resources:** Hana Krismawati, Arry Pongtiku, Hong Liu, Furen Zhang.

**Supervision:** Antonius Oktavian.

**Validation:** Hana Krismawati, Astrid Irwanto, Ishak Darryl Irwan.

**Writing – original draft:** Hana Krismawati, Astrid Irwanto.

**Writing – review & editing:** Hana Krismawati, Astrid Irwanto, Yonghu Sun, Hong Liu, Antonius Oktavian, Jianjun Liu.

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
