## [Decision Letter · Decision Letter 0]

23 Jul 2020

Dear Dr. Liu,

Thank you very much for submitting your manuscript "Validation study of HLA-B*13:01 as a biomarker of dapsone hypersensitivity syndrome in leprosy patients in Indonesia" for consideration at PLOS Neglected Tropical Diseases. As with all papers reviewed by the journal, your manuscript was reviewed by members of the editorial board and by several independent reviewers. The reviewers appreciated the attention to an important topic. Based on the reviews, we are likely to accept this manuscript for publication, providing that you modify the manuscript according to the review recommendations. 

Sincerely,

Johan Van Weyenbergh

Associate Editor

Ana LTO Nascimento

Deputy Editor

Reviewer's Responses 

Reviewer #2: The paper “Validation study of HLA-B*13:01 as a biomarker of dapsone hypersensitivity syndrome in leprosy patients in Indonesia” Krismawat et al evaluates the allele HLA-B*13:01 association with dapsone hypersensitivity among Indonesians. This data is not original but an important replication of previously described among Chinese Thai and other populations. The paper is well written and aims are correctly addressed. The sample size is small but due to the high odds ratios expected, calculations supported the numbers with good power estimations. The genotyping methods for HLA results were performed adequately and logistic regression is the best analysis approach to test association.

Reviewer #2: The findings are important and deserve publication. The results are very clear and provides good evidence to screen the Indonesian population prior to dapsone treatment avoiding life-threatening conditions for leprosy patients.

Reviewer #2: Authors must discuss a little better the introduction of this screening as a policy and a great improvement would be cost-effectiveness analysis to evaluate impact in incidence and mortality. Indeed, a recent publication listed by authors performed a good prospective study and authors cited as “Screening for this allele in new leprosy patients before the initiation of MDT has proven beneficial to reduce incidence rate of DHS in China to zero”. Authors should consider follow up and complementary to this previous analysis, with cost-effectiveness analysis.

Reviewer #1: Major and general comments: This is an important area and there are potentially interesting findings here in order to draw appropriate conclusions from the data the analysis should be redone and that much more of the actual data including the causality assessment and functional data should be included for the readers (supplement).

 In terms of the analysis at a minimum this should include matching cases to age,sex, race and match for underlying disease state and co-morbidities and conditional logistical analyses. functional data and matched analysis would be enormously helpful to clean this up so conclusions can be drawn from the data.

With regards to previous study showing an association between HLA-B*13:01 has been described in association with co-trimoxazole-induced DRESS (PMID: 32452529), do you have cases or control using co-trimoxazole? It should be discussed cross reactivity between Dapsone and co-trimoxazole.

The previous study showed that HLA-B*13:01 was the predictive markers for both DRESS and SJS-TEN. In this study the authors should reanalyze to confirm the phenomenon.

As too many HLA-B genotypes were discovered and compared, the p-value should be corrected by Bonferroni correction.

Table 1 showed and compared alleles frequency of HLA-B genotype. As we known that carry only one allele can be the risk individual, the number of carrier should be expressed and analyzed. 

The general population from the same race should be recruited and genotyped to compare and it can help to extrapolate for screening in clinical implementations. 

HWE was less than 0.05, the authors should describe and discuss. 

The clinical characteristics for cases-controls should be draw out. Recently, the underlying condition such as renal function can determine the risk of drug-induced SCARs such as allopurinol. Dose can also the risk factors. The authors should reanalyze the non-genetics risk factors. 

The authors should show number need to test to determine how many patients would be needed to screen to identify one patient likely to go onto DHS.

Please check through the manuscript for genetic format, it should be italic form.

The first paragraph of Introduction (line 62-65), second paragraph (line 75-78) need reference. 

As the authors mentioned that the leprosy patients got multi-drug treatment, how can the culprit Dapsone as drug-induced hypersensitivity, the Elispot and LTT should be performed to confirm the DHS.

 What is the criteria to define DHS.

The causality assessment should be performed and express in table format.
---

## [Editor Report · Decision Letter 1]

25 Aug 2020

Dear Dr. Liu,

We are pleased to inform you that your manuscript 'Validation study of HLA-B*13:01 as a biomarker of dapsone hypersensitivity syndrome in leprosy patients in Indonesia' has been provisionally accepted for publication in PLOS Neglected Tropical Diseases.

Best regards,

Johan Van Weyenbergh

Associate Editor

Ana LTO Nascimento

Deputy Editor

---

## [Editor Report · Acceptance letter]

8 Oct 2020

Dear Dr. Liu,

We are delighted to inform you that your manuscript, "Validation study of *HLA-B*13:01* as a biomarker of dapsone hypersensitivity syndrome in leprosy patients in Indonesia," has been formally accepted for publication in PLOS Neglected Tropical Diseases.

Best regards,

Shaden Kamhawi

co-Editor-in-Chief

Paul Brindley

co-Editor-in-Chief
